# Automatic Extraction of Supraglacial Lakes in Southwest Greenland during the 2014–2018 Melt Seasons Based on Convolutional Neural Network

**Jiawei Yuan** [1,2,4,7], **Zhaohui Chi** [3], **Xiao Cheng** [1,2,4,7,*], **Tao Zhang** [5], **Tian Li** [6] and **Zhuoqi Chen** [1,2,4,7]

1   State Key Laboratory of Remote Sensing Science, College of Global Change and Earth System Science, Beijing Normal University, Beijing 100875, China; jwyuan@mail.bnu.edu.cn (J.Y.); chenzq_2019@163.com (Z.C.)
2   School of Geospatial Engineering and Science, Sun Yat-Sen University, Guangzhou 519082, China
3   Department of Geography, Texas A&M University, College Station, TX 77843, USA; zchi@tamu.edu
4   Southern Marine Science and Engineering Guangdong Laboratory, School of Marine Sciences, Sun Yat-Sen University, Guangzhou 519082, China
5   State Key Laboratory of Remote Sensing Science, Jointly Sponsored by Beijing Normal University and Institute of Remote Sensing and Digital Earth of Chinese Academy of Sciences, Beijing 100875, China; 201721480054@mail.bnu.edu.cn
6   School of Geographical Sciences, University of Bristol, Bristol BS8 1QU, UK; tian.li@bristol.ac.uk
7   University Corporation for Polar Research, Beijing 100875, China
*   Correspondence: chengxiao9@mail.sysu.edu.cn

**Abstract:** The mass loss of the Greenland Ice Sheet (GrIS) has implications for global sea level rise, and surface meltwater is an important factor that affects the mass balance. Supraglacial lakes (SGLs), which are representative and identifiable hydrologic features of surface meltwater on GrIS, are a means of assessing surface ablation temporally and spatially. In this study, we have developed a robust method to automatically extract SGLs by testing the widely distributed SGLs area—in southwest Greenland (68°00′ N–70°00′ N, 48°00′ W–51°30′ W), and documented their dynamics from 2014 to 2018 using Landsat 8 OLI images. This method identifies water using Convolutional Neural Networks (CNN) and then extracts SGLs with morphological and geometrical algorithms. CNN combines spectral and spatial features and shows better water identification results than the widely used adaptive thresholding method (Otsu), and two machine learning methods (Random Forests (RF) and Support Vector Machine (SVM)). Our results show that the total SGLs area varied between 158 and 393 km$^2$ during 2014 to 2018; the area increased from 2014 to 2015, then decreased and reached the lowest point (158.73 km$^2$) in 2018, when the most limited surface melting was observed. SGLs were most active during the melt season in 2015 with a quantity of 700 and a total area of 393.36 km$^2$. The largest individual lake developed in 2016, with an area of 9.30 km$^2$. As for the elevation, SGLs were most active in the area, with the elevation ranging from 1000 to 1500 m above sea level, and SGLs in 2016 were distributed at higher elevations than in other years. Our work proposes a method to extract SGLs accurately and efficiently. More importantly, this study is expected to provide data support to other studies monitoring the surface hydrological system and mass balance of the GrIS.

**Keywords:** supraglacial lakes; Convolutional Neural Networks (CNN); Landsat 8 OLI images; morphological and geometrical algorithms; change detection

## 1. Introduction

The Greenland Ice Sheet (GrIS) is the second largest ice sheet in the world after the Antarctic Ice Sheet. GrIS could cause a global sea level rise of 7 m if it melts completely, with the uncertainty at 35% [1]. GrIS has shown accelerated mass loss and contribution to sea level rise in recent decades due to global warming [2,3]. GrIS has made a total contribution of 13.7 ± 1.1 mm to the global sea level rise since 1972 [4]. Mass loss of GrIS mainly results from a combination of surface runoff, submarine melt, and iceberg calving [5–8]. Surface runoff led to nearly 40% of GrIS's mass loss from 2000 to 2008 [5]. Increased surface melting and runoff has contributed to 84% of the increase in mass loss from 2009 to 2012 [9]. Therefore, surface runoff of GrIS has currently drawn attention in the relevant studies [10].

In general, GrIS's surface meltwater, mainly caused by solar radiation, is part of supraglacial hydrological system. The system is mainly located in the ablation area and includes Supraglacial lakes (SGLs), streams, crevasses, and moulins [11]. Meltwater accumulates in SGLs and is transported through streams and crevasses; some goes directly across the GrIS surface to the ocean [12], while most flows through vertical surface-to-bed hydrologic connections (moulins) and finally reaches the subglacial area [13] and then the ocean. The influence of GrIS surface meltwater on the ice sheet mass balance needs to be understood, and an important step is to quantify the surface meltwater in SGLs. As the most representative and identifiable hydrologic feature, SGLs have been investigated based on geomorphological characterizations such as quantity, area, depth, and volume [11]. Besides these physical properties' extraction, it has also been found that SGLs contribute to the negative mass balance of GrIS in two main ways. First, the melting rate of ice sheets is affected by the low albedos of SGLs, which cause increased solar radiation absorption [14–16]. Second, SGLs influence the ice sheet dynamic process by transporting surface meltwater to the GrIS bed, reducing basal friction, and thereby accelerating ice sheet motion in early-melt season [17–25], although this is followed by deceleration in marginal regions [26].

Different remote sensing images and threshold methods have been used in previous studies of SGLs. MODIS images are mainly used in long time series to track the evolution of SGLs because of the long time coverage and high temporal resolution. Sundal et al. [27] found the monthly area variations of SGLs at different elevations using 260 MODIS images acquired between 2003 and 2007, based on setting the threshold of ratio between red and blue bands. Selmes et al. [13] used 3704 MODIS images acquired from 2005 to 2009 to map the evolution of 2600 SGLs for the whole GrIS by the moving window thresholding method. However, compared with the low spatial resolution of MODIS images, Landsat images have a higher spatial resolution and more spectral information, so they are more capable of extracting the area and drainage events of SGLs in detail. Miles et al. [28] used a combined Landsat 8 and Sentinel-1 dataset and the threshold of NDWI (Normalized Difference Water Index) to extract the drainage events of SGLs from June to September 2015. Williamson et al. [29,30] designed two algorithms named "FAST" and "FASTER" to extract the area and volume of SGLs based on thresholds of the ratio between red and blue and NDWI, respectively. Schwatke et al. [31] used five water indexes MNDWI (Modified Normalized Difference Water Index), NWI (New Water Index), $AWEI_{nsh}$ (Automated Water Extraction Index for Non-Shadow Areas), $AWEI_{sh}$ (Automated Water Extraction Index for Shadow Areas) and $TC_{wet}$ (Tasseled Cap for Wetness) to combine their individual strengths for the water identification in the summer of 2018.

Overall, previous water identification methods are mainly based on setting the threshold of different water feature indices and decision rules. They extract objects by designing effective or representative features, which are called feature engineering. Feature engineering relies on empirical knowledge and there can be a lack of transferability when the study area changes. On the other hand, during the process of feature learning, the input samples are labeled and then the relationship between the input and the label is established in the model. Therefore, the model can actively learn the features of the samples, without defining what exactly the features are [32]. Feature learning is a black-box operation learning rules and patterns from sample data [33]. Although it lacks the explanation of physical processes and principles, it avoids manually designing features and does not

depend on empirical knowledge, so the adaptability is improved. As one of the most widely used feature learning method, convolutional neural network (CNN) which combines spectral and spatial features has achieved success [34–37] in the recognition of different objects including handwritten words [38], human faces [39], buildings [40], and so on. These researches prompted us to explore the water identification of SGLs using CNN due to its capacity for feature learning.

In this study, we aim to provide an efficient method of extracting SGLs using the established CNN framework and morphological and geometrical operations based on Landsat 8 images during the melt seasons of 2014–2018. Moreover, we assess the water identification results of the CNN by comparing them with those of the widely used threshold method (Otsu) and two other machine learning algorithms (RF and SVM). Otsu uses the global threshold for the water highlighting index NDWI to identify water. RF and SVM can learn the water characteristics and classification rules for water recognition based on the spectral information of sample pixels. The comparison results show that CNN has the best water identification ability of these four methods in this study area.

## 2. Study Region and Data

### 2.1. Study Region

The study region (68°00′ N–70°00′ N, 48°00′ W–51°30′ W) covers ~31,000 km$^2$ in southwest Greenland (shown in Figure 1). It is a representative area of flowing outlet glaciers, ice streams, and hydrological systems inclusive of the active appearance of SGLs and has been studied by many previous researchers [29,30,41].

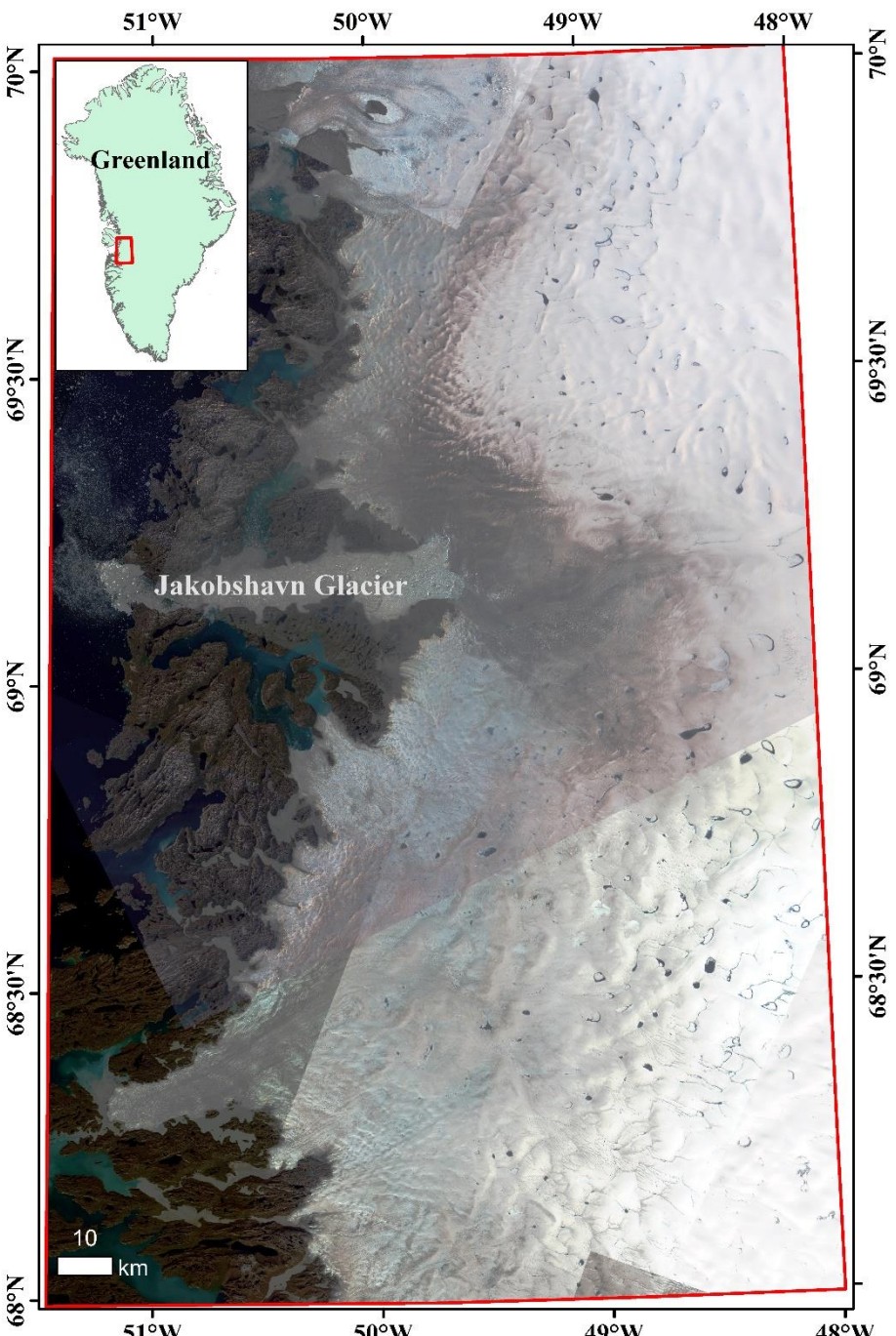

**Figure 1.** The geographic location of the study region in Greenland. The background image is a Landsat 8 30-m true-color average performance image of the melt season of 2018.

## 2.2. Landsat 8 Imagery Collection and Pre-Processing

This study used Landsat 8 images acquired during melt seasons (1 May to 30 September [41]) from 2014 to 2018 and downloaded from Google Earth Engine (GEE) [42]. GEE is a platform provided by Google for online visualization, analysis, and processing of multi-petabyte catalog of satellite imagery and geospatial datasets. It enabled us to perform the following image selecting and processing.

Water shows high reflectance in the visible range between 0.4 µm and 0.7 µm, absorbing nearly all the electromagnetic energy in the near-infrared and shortwave infrared ranges between 0.7 µm and 2.5 µm. Therefore, we used B1–B8 to represent the spectral characteristics of water in our study.

First we selected the images with cloud cover below 1% to exclude the obscuring influence on water recognition. Then we narrowed the time window down to 1 May to 30 September to ensure sufficient representation of SGLs in melt seasons. There are 20–30 images each year that meet these filtration requirements. Finally, we calculated the mean image of these 20—30 images from the five months, inclusive of the average performance of SGLs during the melt season every year.

### 2.3. Sample Selection

With the purpose of identifying water-of-SGL rather than water-of-ocean, we created water-of-SGL (called "water" in the following) and the background as two groups of samples in this study. For each group of samples every year, the selecting process includes two steps based on RGB images: (1) Manually delineate polygons (Figure 2) in the representative area. More specifically, we created polygons for pure water pixels (approximately 60–70 for each year) in the SGLs area. The SGLs with a wide variety of sizes, depths, and shapes are preferred to include different features. We created background polygons (approximately 40–50 for each year), evenly distributed in the study region, representing different types of objects such as water-of-ocean, ice, rocks, and glacial drifts. There is no necessity for the background polygons to be pure pixels because we regard all these objects as the same sample group. (2) Randomly generate sample points (Figure 3) inside the delineated polygons using the "Create Random Points" tool in ArcGIS 10.2 (ESRI, USA). Here randomness increases the representative nature of the samples for the corresponding objects. Hence, we have 1000 water sample points and 1000 background sample points for each year, which add up to 10,000 sample points for five years (2014–2018). These sample points are the preparation for the inputs of CNN model.

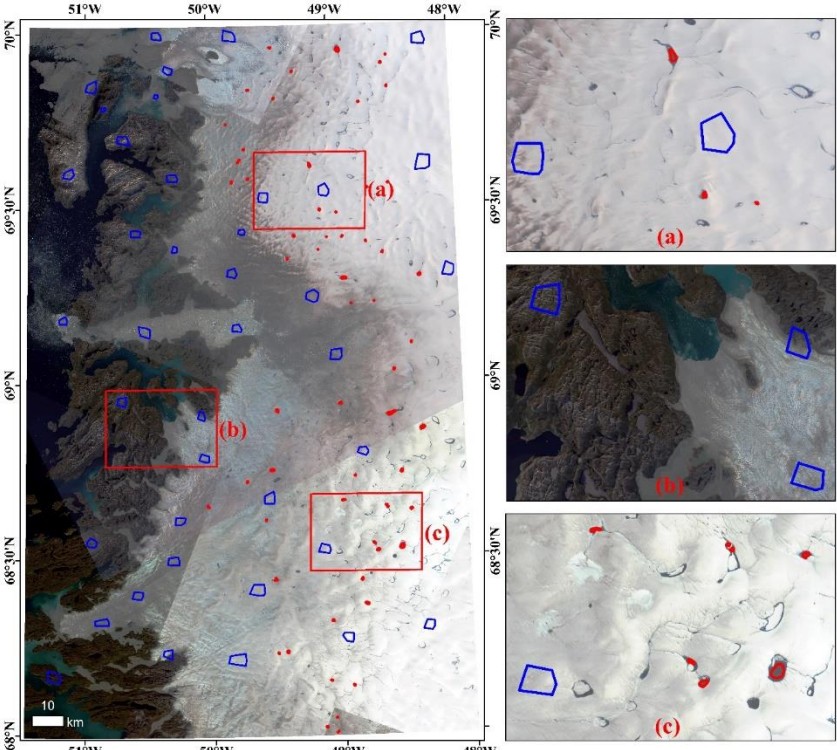

**Figure 2.** Manually delineated polygons within the study area. Red polygons stand for water sample, and the blue polygons stand for background sample. (**a**–**c**) correspond to three detailed images. The background image is the same as Figure 1.

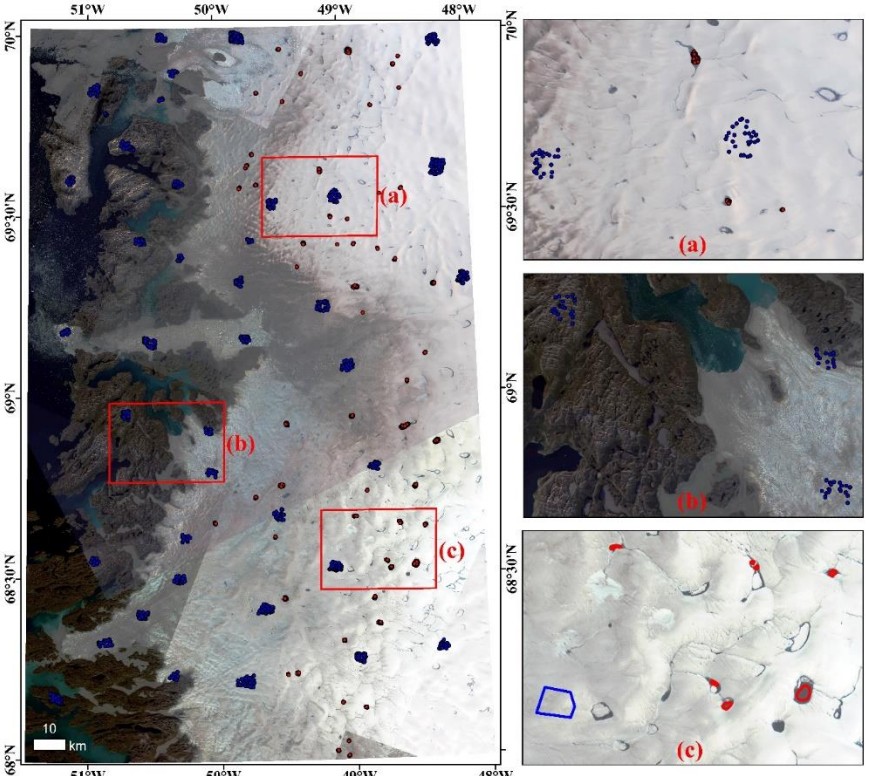

**Figure 3.** Sample points created from the polygons shown in Figure 2. Red points stand for water samples, and blue points stand for background samples. (**a**–**c**) correspond to three detailed images. The background image is the same as Figure 1.

## 3. Methods

We first computed the mean image of each band for each year. Secondly, we selected samples based on 10-band composition mean images of B1–B8 from Landsat 8, the Normalized Difference Vegetation Index (NDVI), and NDWI because all 10 bands include the characteristics of water. Thirdly, the CNN model was trained, validated, and tested by all samples selected from these mean images. Finally, the images acquired from five consecutive years were put into the well-trained CNN model to identify water pixels. Then morphological and geometrical operations were applied to extract SGLs from the water.

We extracted SGLs through two main procedures. As shown in Figure 4, the first step was to identify water pixels by training and testing the CNN model using the selected samples. The second step was to extract the object SGLs in five years based on some filtration conditions including the area and width of the water patches, and mathematical operations of geometry and morphology.

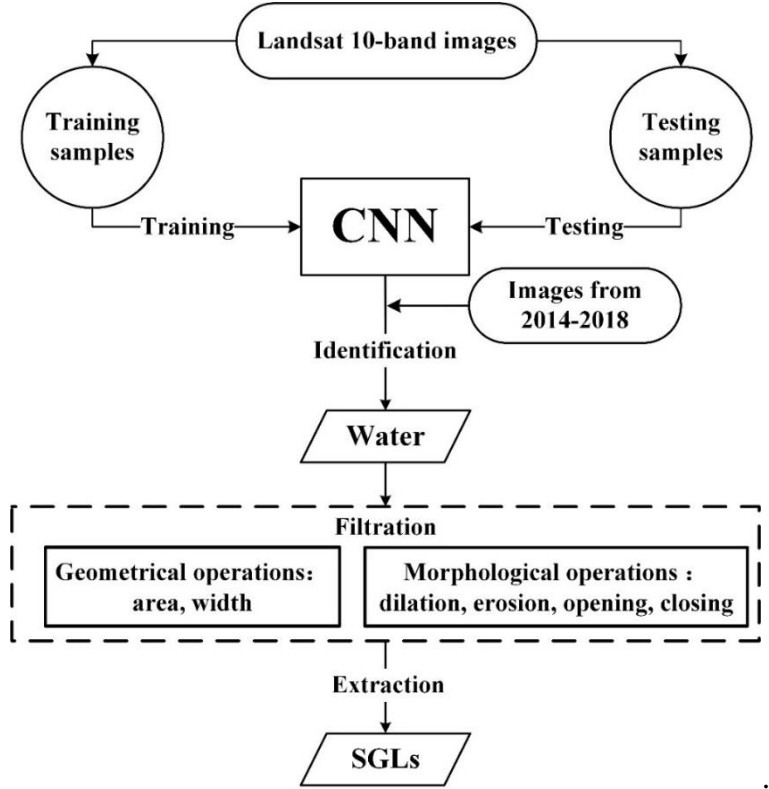

**Figure 4.** Flowchart of supraglacial lakes (SGLs) extraction.

### 3.1. Water Identification Based on CNN

To benefit from the multiple spectral bands in Landsat 8 OLI sensors, many empirical indices have been used to effectively characterize land cover categories by a combination of a subset of Landsat spectral bands such as NDWI [43] and modified NDWI (MNDWI) [44]. Zhang et al. [45] input Landsat 8 image patches with the size of 5 × 5 into the CNN to identify the built-up areas in different cities in China. This inspired us to design the networks as shown in Figure 5, consisting of input layer, convolution layer, fully connected layer and output layer.

The B8 (with an original resolution of 15 m) of the Landsat 8 OLI images was resampled to 30-m resolution using the nearest neighborhood sampling. In order to make full use of the spectral information of the Landsat 8-OLI image, B1–B8 are stacked together with the NDWI images calculated by Equation (1) and the NDVI images calculated by Equation (2). Hence, the 10-band image was taken as input for the CNN. Due to the variability and inhomogeneity of the geometrical shape of surface meltwater, spatial information plays an important role in the accurate identification of water pixels. In the 30-m resolution image, we considered the size of every image patch as N × N (3, 5, 7, 9, and 11 have all been tested as the value of N). For each pixel, an N-neighborhood was considered, which means the size of image patch is N × N × 10. So, a 10-band image patch shown as N × N neighborhood centered on each sample point was inputted into the neural network. The optimal value of N was determined based on the accuracy of the training process. When the training process converged, the corresponding N with the highest accuracy would be the optimal value. There are two convolutional layers and two max-pooling layers that aim to extract spectral features and spatial features and higher-grade semantic features. There are three layers in the fully connected layer. In order to prevent overfitting, a batch-normalization layer and a dropout layer were added. The output layer consists of a softmax operator, which outputs water and background into two separate categories.

In the whole network, we use the activation function called Rectified Linear Unit (ReLU) [46] to solve the vanishing gradient problem for training epochs in the network.

$$NDWI = \frac{B3(Green) - B5(NIR)}{B3(Green) + B5(NIR)} \tag{1}$$

$$NDVI = \frac{B5(NIR) - B4(Red)}{B5(NIR) + B4(Red)} \tag{2}$$

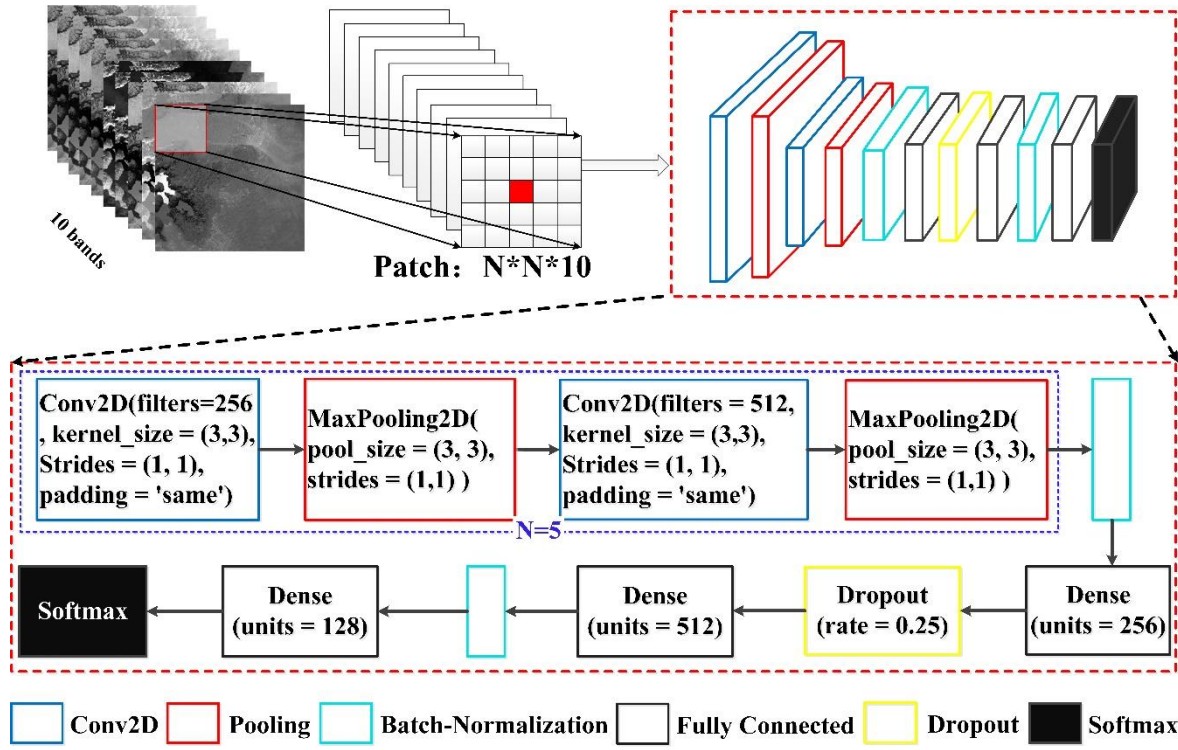

**Figure 5.** Framework structure of the Convolutional Neural Networks (CNN).

For the CNN, we established three hyperparameters to better train the system: batch size, epoch and learning rate. The batch size was set to 128 and the epoch was set to 200. We used cross entropy as the loss function and stochastic gradient descent (SGD) as the optimizer to learn the model parameters. The initial learning rate of SGD was set to 0.001. When the learning rate was no less than 0.00001, the learning rate decreased exponentially with the increase in training epoch indices. The formula for the learning rate is shown in Equation (3). In this equation, 0.001 represents the initial learning rate, *lrate* represents the learning rate after each iteration, and *n* indicates the epoch of iteration.

$$lrate = \begin{cases} 0.00001, & lrate < 0.00001 \\ 0.5^{\frac{1+n}{10}} \times 0.001, & lrate >= 0.00001 \end{cases} \tag{3}$$

The hyperparameters connecting the networks in CNN need to be determined during the training process. Here we used 42%, 18%, and 40% [45] of all the 10,000 samples as the training set, validation set, and testing set, respectively. The training set is used to learn and adjust the weights and offsets in the neural network, while the validation set is used to verify that the hyperparameters are reliable. The training process stops after 200 epochs. Therefore, if the accuracies of training and validation sets are high (greater than 0.9) and close (the difference is smaller than 0.05) and the losses are low (smaller than 0.2), the network is successfully trained, which indicates that the model has converged.

### 3.2. Accuracy Assessment for Water Identification

The test samples were classified into two categories to get the predictive label. A confusion matrix shown in Table 1 was then obtained according to the true label and the predictive label. The overall accuracy (OA), recall, and precision were calculated based on the confusion matrix. OA represents the proportion of correctly predictive samples in all samples. Recall indicates the percentage of predictive water samples in all true water samples. Precision indicates the percentage of true water samples in all predictive water samples. These three precision indices can be used to assess the accuracy of water area identification.

**Table 1.** Confusion matrix for accuracy assessment.

| | | Prediction | | |
|---|---|---|---|---|
| | | **Background** | **Water** | **Sum** |
| **Ground Truth** | **Background** | True Negative (TN) | False Positive (FP) | Actual Negative (TN + FP) |
| | **Water** | False Negative (FN) | True Positive (TP) | Actual Positive (FN + TP) |
| | **Sum** | Predicted Negative (TN + FN) | Predicted Positive (FP + TP) | TN + TP + FN + FP |

The *OA*, *Recall*, and *Precision* are calculated using Equations (4)–(6), respectively:

$$OA = \frac{TP + TN}{TP + TN + FN + FP} \tag{4}$$

$$Recall = \frac{TP}{TP + FN} \tag{5}$$

$$Precision = \frac{TP}{TP + FP} \tag{6}$$

### 3.3. The Extraction of SGLs from Water

SGLs' geometric characteristics such as area, length, width, and shape are very important for SGLs extraction. Therefore, we considered extracting SGLs from the water identification results by using an algorithm for geometry and morphology. There are three conditions that should be taken into consideration in the process of extracting SGLs from water: (1) SGLs smaller than the minimum area threshold should be removed; (2) holes resulting from the floating ice inside the SGL should be filled and the outer contour should represent the size of the SGL; (3) narrow water patches that represent rivers should be removed. The flow chart shown in Figure 6 illustrates how these problems have been solved.

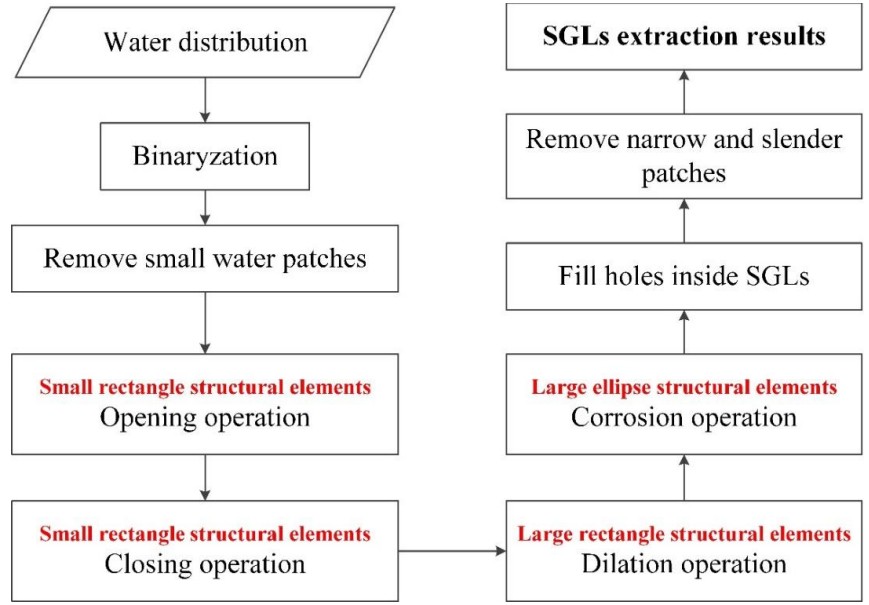

**Figure 6.** Flowchart designed to extract SGLs from water.

First, we converted the water identification image to binary in order to proceed with the morphological and geometrical algorithm. Then, according to the minimum SGL size of two MODIS pixels (250 m) in previous research [13,29], water patches smaller than 200 pixels (~0.18 km$^2$, called tiny water patches) were removed. Next, we conducted opening and closing operations based on small rectangle structural elements. The following Equations (7)–(10) show the opening, closing, dilation, and erosion operations displayed by the symbols ○,●,⊕,⊙, respectively. The opening operation (shown in Equation (7)) means that erosion is done before the dilation to eliminate isolated water spots; it is used to delete edge fragmentation around water patches to form more rounded shapes. Contrarily, the closing operation (shown in Equation (8)) means the dilation is done before the erosion process to connect scattered water patches inside SGLs to fill small holes. After that, we conducted a dilation operation (shown in Equation (9)) based on a large rectangle structural element to connect major SGLs, and we conducted an erosion operation (shown in Equation (10)) based on a large ellipse structural element to make the SGLs more compact. The hole-filling process was iterated until no void appeared in each SGL. To distinguish narrow rivers from SGLs, water patches with a width of the minimum bounding rectangles less than 5 pixels (~150 m, called slender rivers) were regarded as narrow rivers [47] and were removed from the final results. All of these processes were accomplished on a Python platform using two open-source libraries: Opencv and Scikit-image.

The opening operation of structural element B to A:

$$A○B = (A⊙B)⊕B \qquad B = \text{strel('rectangle',5,5)}. \qquad (7)$$

The closing operation of structural element B to A:

$$A●B = (A⊕B)⊙B \qquad B = \text{strel('rectangle',5,5)}. \qquad (8)$$

The dilation operation of structural element B to A:

$$(A⊕B) \qquad B = \text{strel('rectangle',20,20)}. \qquad (9)$$

The erosion operation of structural element B to A:

$$(A⊙B) \qquad B = \text{strel('ellipse',10,10)}. \qquad (10)$$

### 3.4. The Temporal–Spatial Change Detection of SGLs

The change degree of area in temporal scale is a quantitative description of the dynamic changing characteristics of SGLs over the study period [48]. A dynamic variability model [49], shown in Equation (11) is used to describe the rate of change of SGLs' area within a certain time frame in the research area. In this formula, $S_a$ and $S_b$ represent the total area of SGLs in different year "a" (previous) and year "b" (next) respectively. K represents the interannual areal changing rate of SGLs. Positive and negative K values mean increasing and decreasing, respectively, the absolute value of K implies the speed of SGLs' change:

$$K = \frac{S_b - S_a}{S_a} \times 100\% \tag{11}$$

We calculated the difference in lake area on pixel basis between two years. As shown in Equation (12), $D_a$ and $D_b$ represent SGLs' spatial distribution in years "a" (previous) and "b" (next). *CH* represents the result of change detection and has three possible values: 0 shows no change, 1 indicates it changes from background to SGL, and −1 indicates it changes from SGL to background.

$$CH = D_b - D_a \tag{12}$$

### 3.5. Three Algorithms (Otsu, SVM, and RF) Used for Comparison with CNN

We use both unsupervised and supervised algorithms to make well-rounded comparison with CNN. Otsu [50] is an unsupervised algorithm to determine the global threshold for image binary segmentation. It obtains the threshold when the interclass variance between the target and the background image is the largest. This is consistent with our aim to distinguish water from the background. NDWI can highlight the water information in the image with the value range of [−1, 1]. When the value is close to 1, the pixel is water, and when the value is near to −1, the pixel is non-water. Water and non-water can be distinguished by setting an appropriate threshold. The calculation of Otsu is simple and fast, but when the ratio between the target and background is large, the variance of classes displays multiple peaks and the segmentation effect can be inaccurate.

Besides unsupervised algorithms, supervised algorithms should also be included in the comparison. RF and SVM are both classical and widely used supervised machine learning algorithms with the high classification accuracy and efficiency. The main idea of SVM [51] is to establish an optimal decision hyperplane to maximize the distance between the nearest two classes of samples on both sides of the plane. SVM has the advantage of low generalization error rate, but it has difficulty in solving multiple classification problems. The essence of RF [52] belongs to a parallel ensemble classification algorithm, which is an improvement on the decision tree algorithm that merges multiple decision trees together; the establishment of each tree depends on the samples extracted independently. RF is computationally efficient and able to process images with multi-features. However, it could perform overfitting when there is obvious image noise. In this research, we use the two classifiers (svm.SVC and ensemble.RandomForestClassifier) provided by the Python sklearn module. Referring to [45], and through theoretical analysis and experiment, the parameters of the two classifiers are shown in Table 2.

**Table 2.** Parameters of classifiers used in our experiments (Python sklearn).

| Algorithm | Abbreviation | Parameter Type | Parameter Name (sklearn) | Parameter Set |
|---|---|---|---|---|
| **Support Vector Machine** | SVM | kernel type | kernel | rbf |
| | | penalty coefficient | C | 10 |
| | | gamma | gamma | 100 |
| **Random Forests** | RF | base classifier | base_estimator | decision Tree |
| | | number of trees | n_estimators | 60 |

## 4. Results

### 4.1. The Training and Testing Results of CNN

Four training processes with different sizes of image patches are shown in Figure 7. As the size of the image patch increases, the time needed to finish training (when all the accuracies and losses are stable) is approximately 60 s, 80 s, 170 s, and 330 s. Obviously, the computation time is shorter when the sizes are $3 \times 3$ and $5 \times 5$, which implies better efficiency. However, losses are lower when the size of the image patch is $7 \times 7$ and $9 \times 9$, due to the higher numbers of feature dimensions.

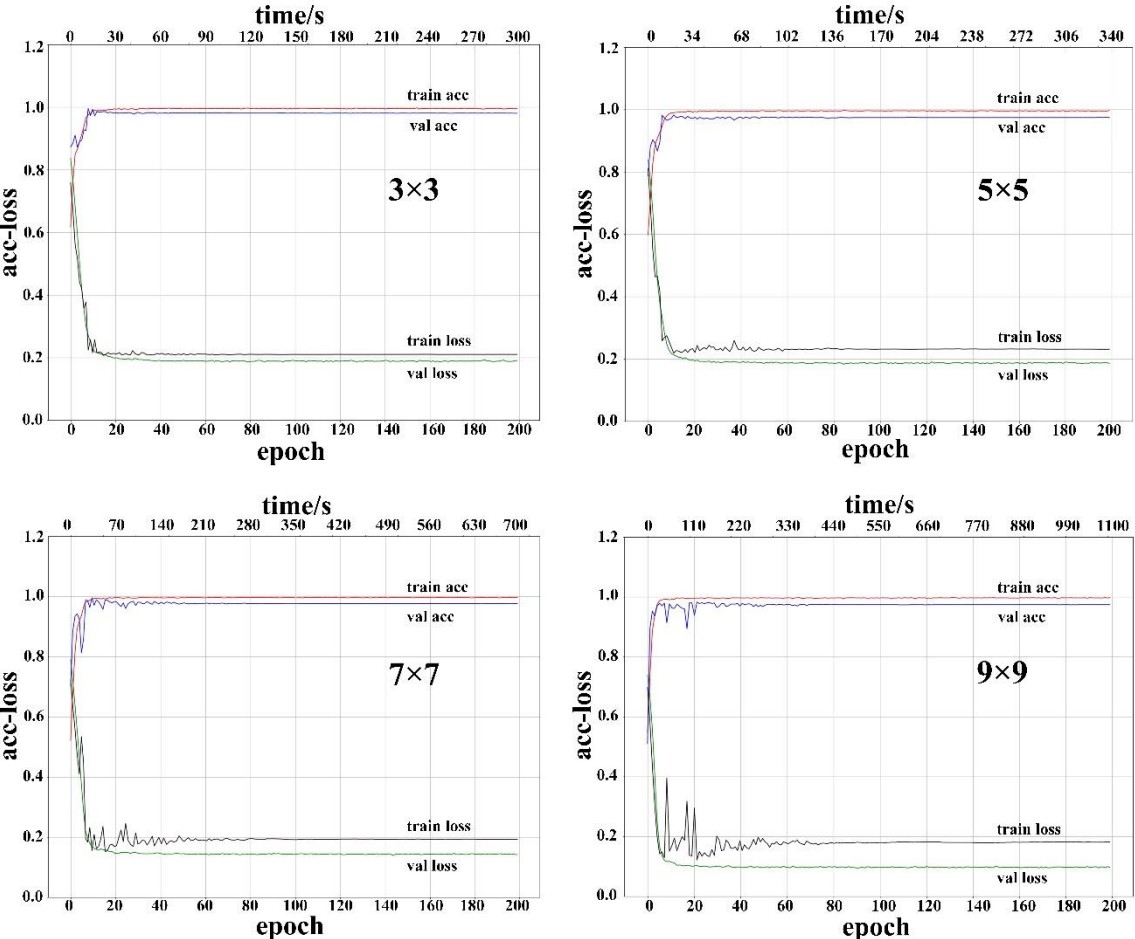

**Figure 7.** Accuracy and loss in the training processes using different sizes of image patches. "Acc" and "val" mean accuracy and validation, respectively.

After getting the four CNN models in the training process, we used a test set to examine overfitting and to determine the optimal size of image patch based on accuracy and time consumption. As shown in Figure 8, comparing the four models with different sizes, all three precision indices are consistently high, and the computation time is relatively short when the size is $5 \times 5$. Therefore, we decided to set the size of image patch to $5 \times 5$ because of the high accuracy and relatively short time consumption.

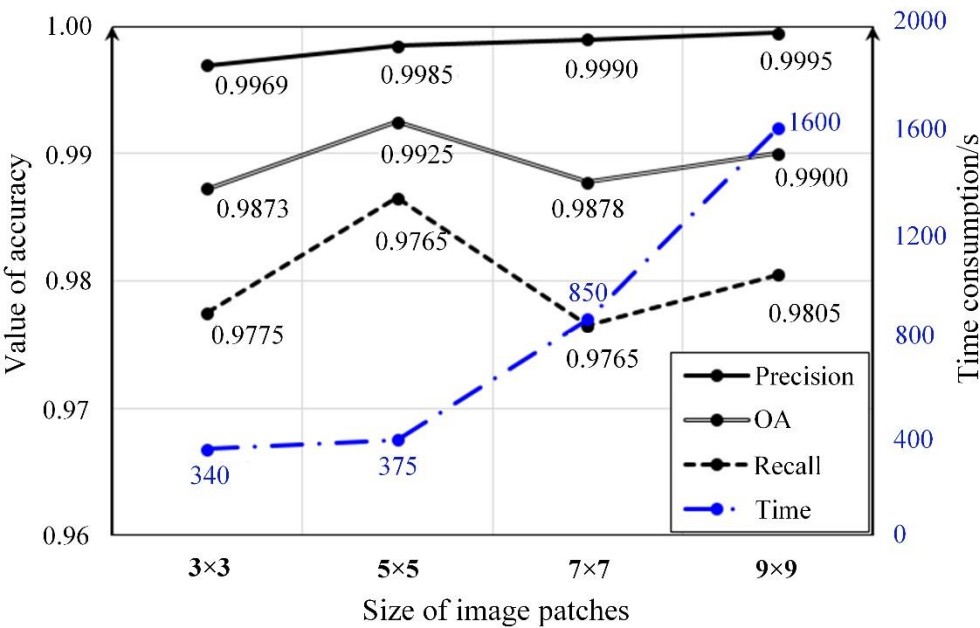

**Figure 8.** Accuracy and time consumption in testing processes using different sizes of image patch.

The detailed information of CNN model with the patch-size of $5 \times 5$ is shown in Table A1, including the layer type, output shape, the number of parameters and the number of neurons in each layer. The "M" in the column of output shape represents the number of samples that are actually trained. There are no parameters in pooling layers, dropout and flatten (converting a multidimensional feature vector into a one-dimensional vector) layers, and no neurons in dropout and batch-normalization layers. Total 482,850 parameters are divided into the 482,338 trainable and the 512 untrainable (the mean and variance of batch-normalization layer).

### 4.2. Accuracy Evaluation

We used the CNN model trained by a $5 \times 5$ image patch to identify water in the melt seasons from 2014 to 2018. The model goes through every center pixel from top left to bottom right in the form of a $5 \times 5$ image patch, then makes decisions about whether it is water or not. As shown in Figure 9, the identified red water pixels cover the real water areas of SGLs, regardless of whether the shapes are irregular circles, elongated circles, narrow segments or hollow rings.

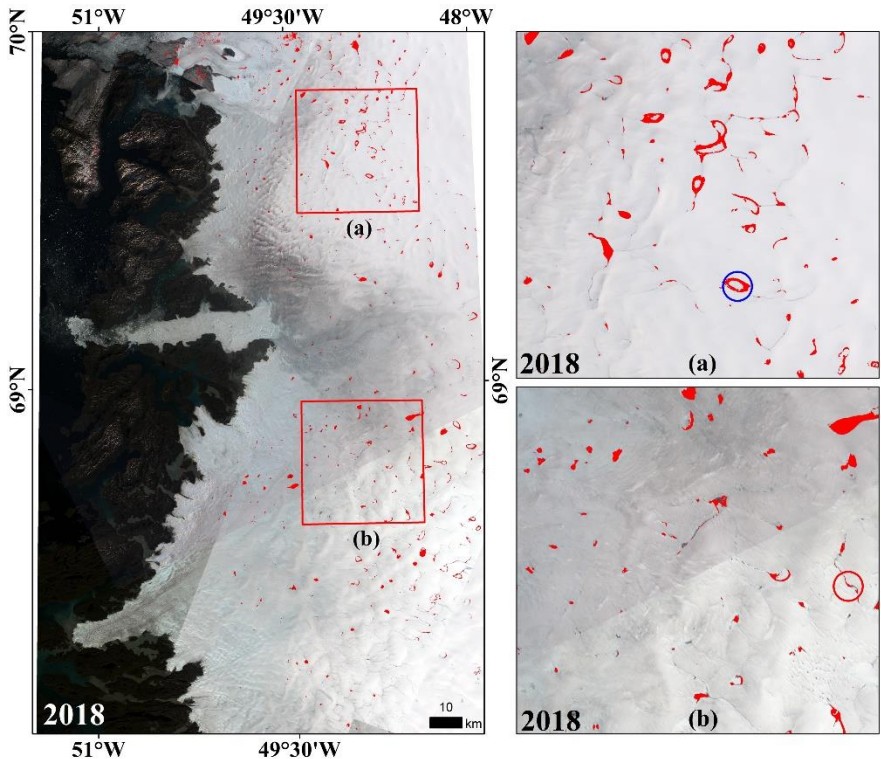

**Figure 9.** Water identification results in 2018. (**a**,**b**) Two detailed subregions from the subfigure on the left. Red stands for identified water. The blue circle shows floating ice in the SGL as a hollow ring and the red circle shows a river as a narrow segment.

Accuracy assessment is based on the testing set with 800 samples from each year studied. As shown in Table 3, (1) the three precision indices are all above 0.94 (with the maximum of 1 and minimum of 0.9475 shown with the numbers in bold), among which OAs are above 0.97, which means the CNN model is well trained and the classification result is reliable. (2) Recalls are above 0.94, so the proportion of predicted water pixels in true water pixels is high, which indicates high accuracy and few omissions. (3) Precision, indicating the proportion of true water pixels in predicted water pixels, is above 0.99. There are nearly no water pixels mixed with the background pixels, particularly for the water-of-ocean area. The high precision indicates a low misjudgment rate. The accuracy in 2018 is relatively low compared to the other years, probably due to the low sample quality. In summary, the overall water identification accuracy for the five years remains at a reliable level.

**Table 3.** Accuracy evaluation indices for water identification processes from 2014 to 2018.

| Year | OA | Recall | Precision |
|------|------|--------|-----------|
| 2014 | 0.9988 | 0.9975 | **1** |
| 2015 | 0.9988 | **1** | 0.9975 |
| 2016 | 0.9925 | 0.9875 | 0.9975 |
| 2017 | 0.9976 | 0.9887 | 0.9999 |
| 2018 | 0.9725 | **0.9475** | 0.9974 |

*4.3. The Extraction Results of SGLs*

We selected five detailed frames from the study area to display SGLs extraction results. As shown in Figure 10, we deleted the false lake areas and extracted the true SGLs' area. Tiny water patches have been deleted. The floating ice, shown as small holes in water identification, has been filled to complete the SGLs. Slender rivers shown as narrow strips have been deleted based on the width threshold.

Although there are a few longer and wider rivers that haven not been deleted, they hold a substantial volume of water so they can be included in the final result of SGL area.

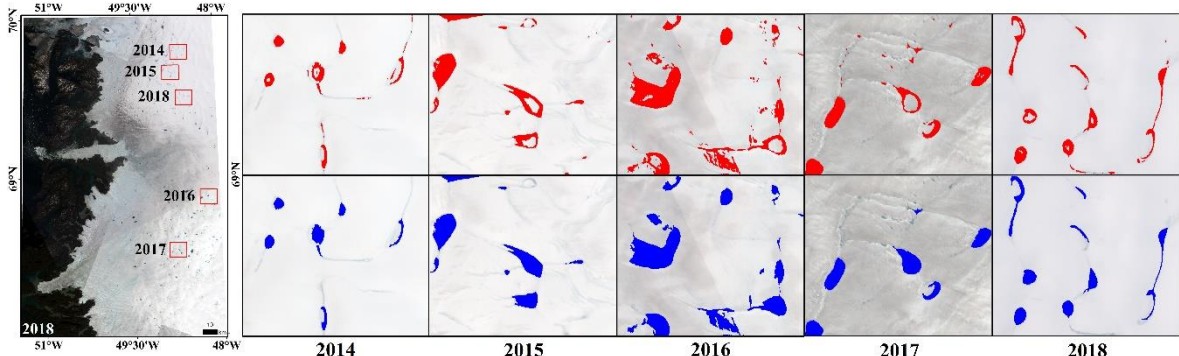

**Figure 10.** The comparison between identified water (**red**) and extracted SGLs (**blue**) during 2014 to 2018 (the left subfigure shows the locations of detailed frames in the five years).

SGLs in southwest Greenland from 2014 to 2018 have been extracted successfully. As shown in Figure 11, the SGLs are mainly distributed in the northern and southern portions of the study region. Their appearances are most active in 2015 and 2016 in the southern and northern areas, respectively, and they both decreased afterwards.

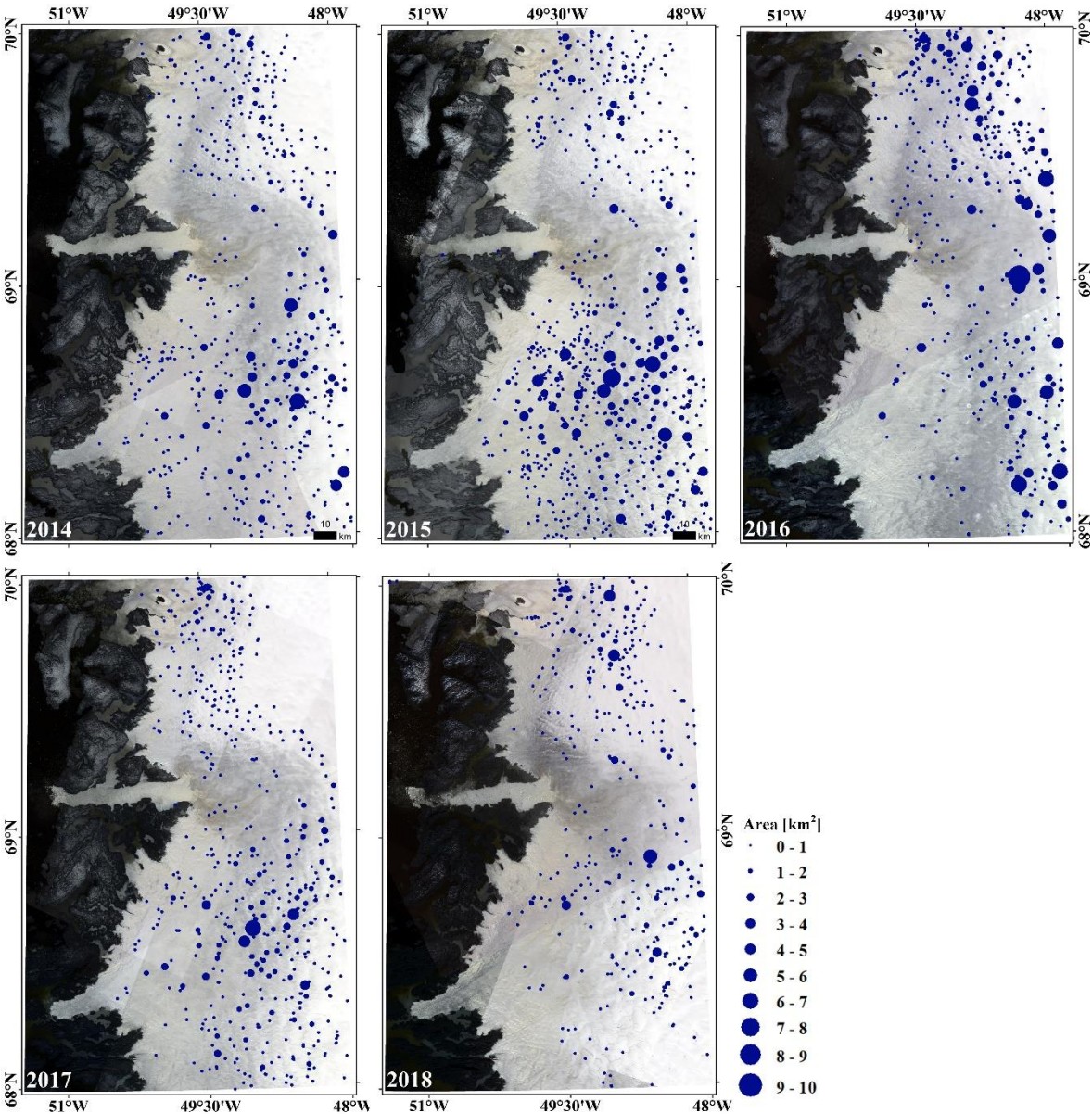

**Figure 11.** SGLs extraction results from 2014 to 2018.

The quantitative descriptive information on the SGLs extracted from 2014 to 2018 is presented in Figure 12. Approximately 400–700 SGLs were observed in the study area during the five years. The maximum individual area ranged from 5.66 to 9.30 km$^2$, and the total area was 158–393 km$^2$. SGLs were most active during the melt season in 2015, with a quantity of 700 and a total area of 393.36 km$^2$.The largest individual lake developed in 2016 with an area of 9.30 km$^2$. This is consistent with the maximum standard deviation, indicating that the SGL area fluctuates the most in 2016. All three attributes showed their lowest values in 2018, which means surface melting was the most limited in that year.

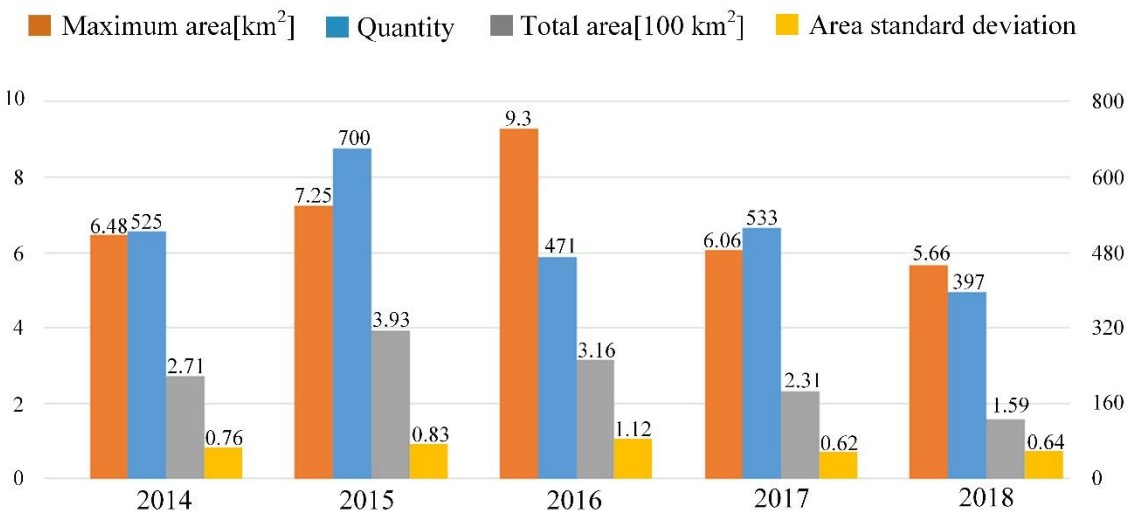

**Figure 12.** The basic information of extracted SGLs from 2014 to 2018.

*4.4. The Dynamic Changes of SGLs from 2014 to 2018*

We calculated the changes of total area and change degrees between every two and three consecutive years to analyze the variation of SGLs in the study area from 2014–2018. It is shown by the value of *CH* in Table 4 that the total area of SGLs had an increase from 2014 to 2015, and kept decreasing over the next three years. K (the interannual change degree of SGLs) was positive from 2014 to 2015, then started to be negative and continued to decrease, reaching a minimum in 2018. Numbers in bold indicate the obvious increase of SGLs area from 2014 to 2015.

**Table 4.** The total area changes of SGLs from 2014 to 2018.

| Time Spans | 2014–2015 | 2015–2016 | 2016–2017 | 2017–2018 |
|---|---|---|---|---|
| *CH* (km$^2$) | **122.45** | −76.97 | −85.81 | −71.85 |
| K (%) | **45.20** | −19.57 | −27.12 | −31.16 |

Spatially, we divided the study area into southern and northern areas at 69° N, and presented the SGLs that had newly appeared (blue), disappeared (orange), and experienced no change (yellow) between two consecutive years. As shown in Figure 13, (1) In the same time scale, the SGLs' change rates in southern and northern areas were the opposite. This means that when SGLs increased in the southern area, they decreased in the northern area, and vice versa. (2) In the same area in different time scales, the trends of SGLs' change are alternating. For the southern area, most SGLs increased, decreased, increased again and decreased again from 2015 to 2018. The changing pattern of the northern area was also alternating, but it was shown as decreasing, increasing, decreasing and increasing respectively. (3) There were 138 SGLs that persisted during all the five years with a combined area of 23.71 km$^2$, and there were 403, 314, 288, 272 SGLs that persisted respectively for two consecutive years from 2014 to 2018.

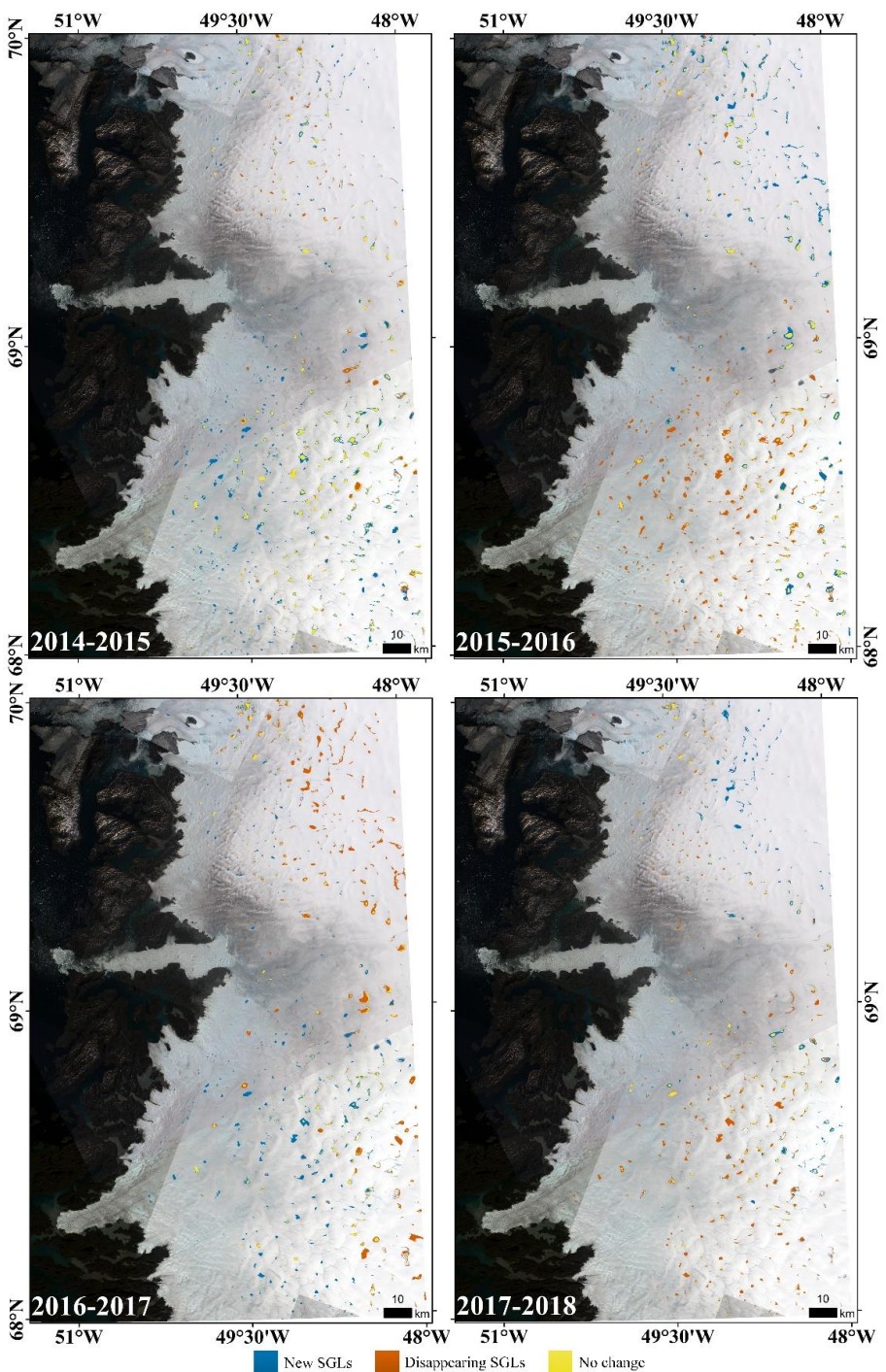

**Figure 13.** Spatial distribution of SGLs' changes between two consecutive years from 2014 to 2018.

We analyzed the elevation distribution and change of SGLs in the consecutive five years by a Digital Elevation Model (DEM). The DEM is from ASTER GDEMV2 with a resolution of 30 m. We calculated the mean elevation of all SGLs based on all pixels in the individual SGL, the elevation of the largest SGL, and the quantity proportions of SGLs in four elevation intervals. As shown with numbers in bold in Table 5, the mean elevation of SGLs and the proportion of SGLs above 1500 m are the highest in 2016, which proves that SGLs are more prevalent at higher elevations in 2016 compared to the other years. Combined with the finding in Figure 12 that the largest SGL appeared in 2016, our results are consistent with the discovery that lakes at higher elevations are also larger because their catchments are greater and the ice is thicker [53].

**Table 5.** The elevation information of SGLs from 2014 to 2018.

| Year | Mean Elevation (m) | Elevation of the Largest SGL(m) | 0–500 m (%) | 500–1000 m (%) | 1000–1500 m (%) | >1500 m (%) |
|---|---|---|---|---|---|---|
| 2014 | 1114 | 1337 | 2.51 | 28.43 | 67.31 | 1.74 |
| 2015 | 1110 | 1146 | 1.17 | 33.92 | 60.79 | 4.11 |
| 2016 | **1193** | 1197 | 2.69 | 22.42 | 66.37 | **8.52** |
| 2017 | 1082 | 1145 | 2.14 | 31.65 | 66.02 | 0.19 |
| 2018 | 1149 | 1210 | 1.31 | 28.72 | 67.36 | 2.61 |

## 5. Discussion

### 5.1. The Comparison of CNN and Otsu for Water Extraction

For this research area, we manually sketch the ice edge line as a mask to remove the non-ice-sheet area where there are several other objects (rock, ocean). This process can avoid multiple peaks in the histogram and control the quantity of objects to be classified as two (water and ice sheet). Then the Otsu algorithm is used to automatically find the global threshold for NDWI, and this threshold segmentation is used to identify water out of ice sheet. Taking 2018 as an example, the comparison of Otsu and CNN water identification results is shown in Figure 14. Subfigure (d) is the histogram of NDWI in the ice sheet area, which shows that there is a large overlap between the two peaks that stand for two objects, and the NDWI value (0.78) between these two peaks is calculated automatically as the threshold by Otsu. As shown in subfigures (b) and (c), Otsu misidentified a large area of ice sheet as water and omitted a large number of water pixels compared with CNN. As for water identification in this ice sheet area, CNN has obvious advantages in terms of its recognition accuracy and water outlining ability.

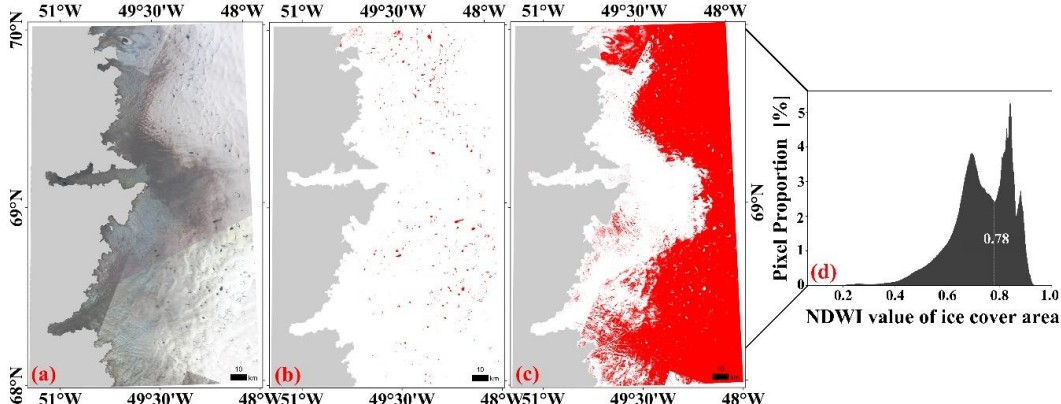

**Figure 14.** Comparison of water identification between CNN and Otsu. (**a**) Landsat 8 30-meter true-color image from 2018; (**b**,**c**) water identification results using CNN and Otsu, respectively. Red stands for identified water and gray stands for non-ice-sheet area. (**d**) Histogram of NDWI from 2018 in the ice sheet area.

### 5.2. The Comparison of CNN, SVM, and RF for Water Extraction

We have selected water-of-SGL (called "water" in the following again) and the background as two groups of samples in Section 2.3. Here the same samples are used to train and test the RF and SVM models. In the training process, the accuracy of the training set is close to 100% when the model converges. In the testing process, the three accuracy indices (OA, Recall, and Precision) of RF and SVM are compared with those of CNN based on the same test samples from 2014 to 2018. As shown in Table 6, the accuracy of CNN for water identification is close to that of RF and SVM, with the maximum difference less than 0.05 (calculated by numbers in bold). However, the accuracy evaluation based on

the limited number of samples may not be sufficient to explain the identification results of all pixels. So, we used the trained RF and SVM model to identify the water in the entire study area, and made comparisons of these three methods.

**Table 6.** Comparison of water identification accuracy among CNN, RF, and SVM.

| Index | Algorithm | 2014 | 2015 | 2016 | 2017 | 2018 |
|---|---|---|---|---|---|---|
| **OA** | CNN | 0.9988 | 0.9988 | 0.9925 | 0.9976 | 0.9725 |
| | RF | 0.9938 | 0.9975 | 0.9863 | 0.9950 | 0.9900 |
| | SVM | 0.9975 | 0.9963 | 1.0 | 0.9988 | 0.9913 |
| **Recall** | CNN | 0.9975 | 1.0 | 0.9875 | 0.9887 | **0.9475** |
| | RF | 0.9875 | 0.9975 | 0.9850 | 0.9900 | 0.9875 |
| | SVM | 0.9950 | 0.9950 | 1.0 | 1.0 | **0.9900** |
| **Precision** | CNN | 1.0 | 0.9975 | 0.9975 | 0.9999 | 0.9974 |
| | RF | 1.0 | 0.9975 | 0.9875 | 1.0 | 0.9923 |
| | SVM | 1.0 | 0.9975 | 1.0 | 0.9975 | 0.9925 |

In Figure 15, there are four detailed subregions showing the water identification results of CNN, RF, and SVM. With the combination of spectral and spatial features, CNN can identify the SGLs' water more accurately, with less salt and pepper noise outside of the SGLs (subregion (1)). What is more, the identified SGLs' water outline from CNN is smoother and more complete, and the omission error of water pixels is lower than RF and SVM (subregions (2) and (3)). Moreover, CNN can distinguish the water-of-SGL from the water in the ice river (subregion (4)) based on the water-of-SGL samples, but RF and SVM misidentify the water in the narrow ice river, which is not SGL. In conclusion, although RF and SVM showed similar accuracy in water classification at the pixel scale, CNN has a better water outlining capability, and includes less noise and fewer omission errors on the object scale for SGL extraction.

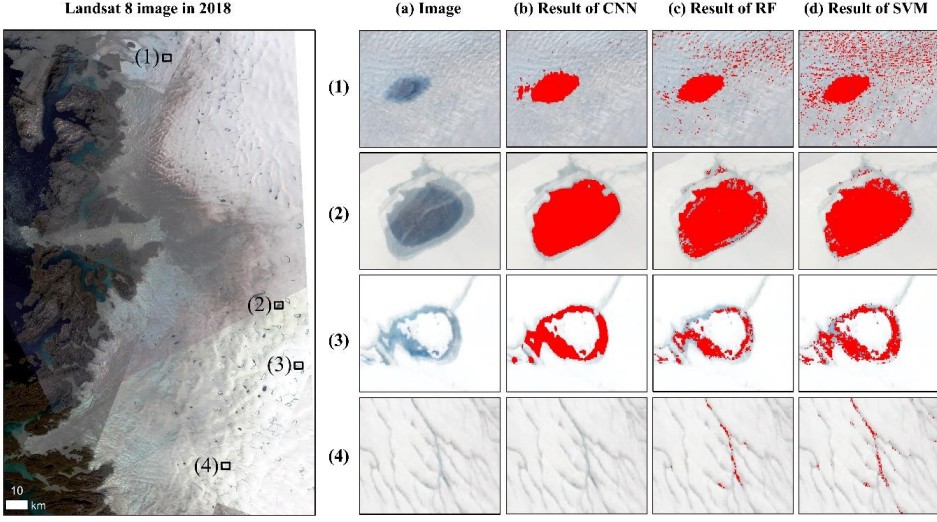

**Figure 15.** Detailed comparison of water identification between CNN, RF, and SVM. (**a–d**) Landsat 8 30-m true-color image from 2018 and water identification results of CNN (without morphological operations), RF, and SVM, respectively. Rows (**1–4**) stand for four detailed subregions from the subfigure on the left. Red stands for the identified water.

### 5.3. The Comparison with the Related Study

We compared our SGL extraction results in 2016 with Williamson et al. [30] with the time period from 1 May to 1 October which is one day longer than ours. We took part of our results that were

located within their smaller study area to make this comparison meaningful. They extracted SGLs with the largest total area of ~130 km$^2$ and the largest individual SGL area of 8.6 km$^2$. These numbers of our results are 153 km$^2$ and 8.2 km$^2$, respectively. The reason why total area in our study is higher may be that the process of calculating mean images can cause co-appearance of SGLs from different periods. As for the largest SGL, due to its appearance in a short time, recording it in one image is easier and more accurate than in a mean image during the entire melt season. So our area of largest SGL is slightly smaller than theirs. In conclusion, the difference of total area is higher than that of the largest SGL's area.

## 6. Conclusions

In this study, we successfully extracted SGLs in southwest Greenland and found the dynamic change characteristics of SGLs from 2014 to 2018. The results show that our method has achieved 94% effectiveness of correctly identifying water, and the accuracy was higher than that achieved by Otsu based on the same samples. What is more, CNN has a better water outlining capability, and includes less noise and fewer omission errors than the other two widely used machine learning methods, RF and SVM. Therefore, CNN combined with the morphological and geometrical algorithms enabled us to extract SGLs automatically and effectively in this study area. The uncertainty of our results is mainly reflected in the total area because of the calculation of mean images during the entire melt season.

Our findings suggest that, in our study region, the total area of SGLs in 2015 (393.36 km$^2$) represented an obvious increase compared to 2014, and kept decreasing after that, reaching a minimum of 158.73 km$^2$ in 2018. There were 138 SGLs with the area of 23.71 km$^2$ existing during all the five years. Seven hundred SGLs were observed in 2015, which was the highest quantity of the five years, while the largest individual lake developed in 2016. There were 175 new SGLs with a total area of 112.45 km$^2$ in 2015, and 303 SGLs with a total area of 234.63 km$^2$ disappeared from 2015 to 2018. In conclusion, the most active surface melting occurred in 2015 and the most limited in 2018.

Spatially, the new SGLs appearance peaked in 2015 in the southern area and 2016 in the northern area. Similarly, many SGLs disappeared in 2016 in the southern area and in 2017 in the northern area. In 2018, the change of SGLs was the most limited. Disappearing SGLs were mainly located in the southern area, and there were a few new SGLs in the northern area. Combining with DEM, we find that SGLs were most active in the area between 1000 and 1500 m, and SGLs in 2016 more appeared at higher elevations than in the other years.

If the spectral information is sufficient and the selected sample are representative and distributed evenly in the study area, the CNN model can be transferred and used in water identification in other regions. In further studies, the transferring and generalization abilities of our CNN model needs to be verified. What is more, we can calculate the volumes of meltwater by combining the SGLs' area from our output and depth from physical algorithm, then assess the mass balance of GrIS. As it has been proven that SGLs form at higher altitudes as temperatures rise [54], and lakes above 1600 m are likely to drain as rapidly as lakes located at lower elevations [55], applying the method we developed in high-temporal-resolution images to explore the relationship between temporal–spatial changes of SGLs and drainage events or meteorological factors is a topic worth investigating.

**Author Contributions:** Conceptualization, J.Y. and X.C.; Data curation, J.Y.; Funding acquisition, X.C. and Z.C. (Zhuoqi Chen); Methodology, T.Z.; Software, T.Z.; Supervision, X.C. and Z.C. (Zhuoqi Chen); Writing—original draft, J.Y.; Writing—review and editing, J.Y., Z.C. (Zhaohui Chi), X.C., and T.L. All authors have read and agreed to the published version of the manuscript.

**Funding:** This research received no external funding.

**Acknowledgments:** This work was supported by the National Natural Science Foundation of China (Grant No. 41925027) and the National Key R&D Program of China (Grant No. 2018YFC1406101).

**Conflicts of Interest:** The authors declare no conflict of interest.

## Appendix A

**Table A1.** The detailed information of CNN model with the size of 5 × 5.

| Layer (type) | Output Shape | Number of Parameters | Number of Neurons |
|---|---|---|---|
| conv2d_1 (Conv2D) | (M, 5, 5, 128) | 11648 | 3200 |
| max_pooling2d_1 (MaxPooling2) | (M, 3, 3, 128) | 0 | 1152 |
| conv2d_2 (Conv2D) | (M, 3, 3, 256) | 295168 | 2304 |
| batch_normalization_1 (Batch) | (M, 3, 3, 256) | 1024 | 0 |
| max_pooling2d_2 (MaxPooling2) | (M, 1, 1, 256) | 0 | 256 |
| flatten_1 (Flatten) | (M, 256) | 0 | 256 |
| dense_1 (Dense) | (M, 128) | 32896 | 128 |
| dropout_1 (Dropout) | (M, 128) | 0 | 0 |
| dense_2 (Dense) | (M, 256) | 33024 | 256 |
| dense_3 (Dense) | (M, 256) | 65792 | 256 |
| dropout_2 (Dropout) | (M, 256) | 0 | 0 |
| dense_4 (Dense) | (M, 128) | 32896 | 128 |
| dense_5 (Dense) | (M, 64) | 8256 | 64 |
| dense_6 (Dense) | (M, 32) | 2080 | 32 |
| dense_7 (Dense) | (M, 2) | 66 | 2 |

Total parameters: 482850, trainable parameters: 482338, non-trainable parameters: 512, total neurons: 8034.

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
