# Peer review of "Automatic Extraction of Supraglacial Lakes in Southwest Greenland during the 2014–2018 Melt Seasons Based on Convolutional Neural Network"

_water, doi:10.3390/w12030891_

Round 1

Reviewer 1 Report

The paper is much improved with improvements both in method description, comparison with other researchers results (section 5.3) and in visual depiction of results (Fig. 11). The main items identified below are simply changes in wording for clarity. Line 352 would benefit by comparison to other results to indicate 2018 was lowest melt year in this region.

23: “on GrIS, are a means of assessing surface ablation temporally and spatially.”

36: Remove “mostly”

72:”compared”

90: reference?

131: Is this more of a training process than a selection process? If so use that as your description.

139: “…randomness increases the representative nature of the samples for the corresponding objects.”

196: “…we established three hyperparameters to better train the system: batch size, epoch and learning rate”

346: Figure 11 is now a much better visualization of SGL size and distribution.

352: “..which suggests surface melting was the most limited in that year.” What does the MAR or other models indicate?

367:  You noted inverse relationship between north and south sectors in previous bullet.  Here you suggest they are the same, need to clarfify. “Taking the southern area as an example, most SGLs increased, decreased, increased again and decreased again from 2015 to 2018, and a similar situation was true in the northern area.”

369: “There were 138 SGL’s that persisted during all five years with a combined area of 23.71km2 and there were 403, 314, 288, 272 SGLs that persisted respectively for two consecutive years from 2014 to 2018.”

380: “…proves that SGLs are more prevalent at higher elevations in 2016 compared to the other years.”

384: Table 5 elevation columns eliminate the elevation decimals the nearest meter is more appropriate.

434: Thank you for the good comparison with Williamson et al (2018)

461: “Spatially, there new SGLs appearance peaked in 2015in the southern area and 2016 in the northern area. Similarly many SGLs disappeared in 2016 in the southern area and in 2017 in the northern areas.”

465: Replace “mostly” with “more”

Reviewer 2 Report

I suggest in the next studies a full adaptation of the model into GEE, the use of CNN ca be successfully integrated as a fully integrated tool for detection of supraglacial lakes

Reviewer 3 Report

I appreciate the Editor to give me a chance to review an interesting and valuable paper. I found some merits in the both methodology and results. In my opinion, this paper has a good potential to be published in the journal. However, I have also some concerns on the different parts of the manuscript. If the author(s) address carefully to the comments, I’ll recommend publication of the manuscript in the journal:

  • Discuss the advantages/disadvantages of three algorithms (Otsu, SVM, and RF) used for comparison with CNN. Why these three? Are they the best selection? Why?
  • Focus on the anomalies of the SGLs extraction results from 2014 to 2018.
  • Add standard deviations for the basic information of extracted SGLs from 2014 to 2018.
  • In the Tables, highlight values that are more important and discuss them for better understanding readers.
  • What are the strategies/recommendations to reduce uncertainties in this study?
  • How can extend the results in other regions with similar/different conditions?
  • It is necessary to characterize a number of layers, neurons on the input/hidden/output layers, training algorithm, activation function, learning rate, momentum, number of epochs, etc. in a distinguished table and discuss on the best structure of the ANN to utilize the results in future works.
  • At the end of the manuscript, explain the implications and future works considering the outputs of current study.
  • The quality of the language needs to improve by a native English speaker for grammatically style and word use.

Round 2

Reviewer 3 Report

I appreciate the authors to address the comments. The quality of the manuscript is acceptable now.